# Cytotoxic Fractions from *Hechtia glomerata* Extracts and *p*-Coumaric Acid as MAPK Inhibitors

**DOI:** 10.3390/molecules26041096

**Published:** 2021-02-19

**Authors:** Tommaso Stefani, Antonio Romo-Mancillas, Juan J. J. Carrizales-Castillo, Eder Arredondo-Espinoza, Karla Ramírez-Estrada, Victor M. Alcantar-Rosales, Leticia González-Maya, Jessica Nayelli Sánchez-Carranza, Isaías Balderas-Renterías, María del Rayo Camacho-Corona

**Affiliations:** 1Facultad de Ciencias Químicas, Universidad Autónoma de Nuevo León, Av. Universidad S/N, Ciudad Universitaria, San Nicolás de los Garza, NL 66451, Mexico; juanjcarrizales@hotmail.com (J.J.J.C.-C.); eder.arredondosp@uanl.edu.mx (E.A.-E.); karla.ramirezst@uanl.edu.mx (K.R.-E.); isaias.balderasrn@uanl.edu.mx (I.B.-R.); 2Laboratorio de Diseño Asistido por Computadora y Síntesis de Fármacos, Facultad de Quimica, Universidad Autónoma de Querétaro, Centro Universitario, Cerro de las Campanas S/N, Santiago de Querétaro, QT 76010, Mexico; ruben.romo@uaq.mx; 3Centro de Investigación y Asistencia en Tecnología y Diseño del Estado de Jalisco, A.C. Servicios Analíticos, Sede Noreste, Parque de Investigación e Innovación Tecnológica, Vía de la Innovación 404, Apodaca, NL 66628, Mexico; valcantar@ciatej.mx; 4Facultad de Farmacia, Universidad Autónoma del Estado de Morelos, Av. Universidad 1001, Col. Chamilpa, Cuernavaca, MO 62209, Mexico; letymaya@uaem.mx (L.G.-M.); jes_chazaq@hotmail.com (J.N.S.-C.)

**Keywords:** *Hechtia glomerata*, cancer, bioassay-guided fractionation, *p*-coumaric acid, kinase inhibitor

## Abstract

Preliminary bioassay-guided fractionation was performed to identify cytotoxic compounds from *Hechtia glomerata*, a plant that is used in Mexican ethnomedicine. Organic and aqueous extracts were prepared from *H. glomerata*’s leaves and evaluated against two cancer cell lines. The CHCl_3_/MeOH (1:1) active extract was fractionated, and the resulting fractions were assayed against prostate adenocarcinoma PC3 and breast adenocarcinoma MCF7 cell lines. Active fraction **4** was further analyzed by high-performance liquid chromatography–quadrupole time-of-flight–mass spectrometry analysis to identify its active constituents. Among the compounds that were responsible for the cytotoxic effects of this fraction were flavonoids, phenolic acids, and aromatic compounds, of which *p*-coumaric acid (*p*-CA) and its derivatives were abundant. To understand the mechanisms that underlie *p*-CA cytotoxicity, a microarray assay was performed on PC3 cells that were treated or not with this compound. The results showed that mitogen-activated protein kinases (MAPKs) that regulate many cancer-related pathways were targeted by *p*-CA, which could be related to the reported effects of reactive oxygen species (ROS). A molecular docking study of *p*-CA showed that this phenolic acid targeted these protein active sites (MAPK8 and Serine/Threonine protein kinase 3) at the same binding site as their inhibitors. Thus, we hypothesize that *p*-CA produces ROS, directly affects the MAPK signaling pathway, and consequently causes apoptosis, among other effects. Additionally, *p*-CA could be used as a platform for the design of new MAPK inhibitors and re-sensitizing agents for resistant cancers.

## 1. Introduction

Among the largest health threats of our time is cancer. It is usually deadly because of innate or acquired resistance to chemotherapy. An estimated 17 million new cancer cases were diagnosed in 2018 [1]. Medicinal plants have always been a source of complex and bioactive metabolites that can be used as therapeutic drugs [2]. *Hechtia glomerata* Zucc. is reported by local people in Mexico to be used in ethnomedicine in various Mexican states to treat respiratory and urogenital infections [3,4]. Furthermore, *H. glomerata* is taxonomically related to the genus *Tillandsia* (Bromeliaceae). Studies reported that a chloroform extract of Jamaican Ball Moss (*Tillandsia recurvata* L.) had cytotoxic activity against three human cancer cell lines (A375 [human melanoma], MCF7 [human breast cancer], and PC3 [human prostate cancer]) and anticancer activity [5,6,7,8]. Based on guidelines for plant selection that were reported by Cos et al. [9] and an oral communication with Prof. Antonio Treviño Rivero (coordinator of the Natural and Traditional Health Program of Universidad Autónoma Agraria Antonio Narro) about ethnomedical use of the plant against cancer in the state of Coahuila, we sought to identify cytotoxic compounds in *H. glomerata* and determine the mode of action of one cytotoxic compound that was isolated and characterized from the most active fraction using microarrays and molecular docking assays.

In our previous studies [10,11], the chemical composition of *H. glomerata* was determined by gas chromatography-mass spectrometry (GC-MS) and ultra-performance liquid chromatography quadrupole time-of-flight mass spectrometry (UPLC-QTOF-MS) analyses of hexane, CHCl_3_/MeOH, and aqueous crude extracts. We also evaluated activity of the extracts and pure compounds against resistant bacteria. The identification of crude extract components was performed by extracting molecular features from the obtained total ion chromatograms (TICs) and then comparing these data with plant-based databases (e.g., PhytoHub) and generic databases (e.g., METLIN) [11]. Furthermore, *p*-coumaric acid (*p*-CA) was isolated from the CHCl_3_/MeOH extract of *H. glomerata* as one of the main constituents of active fractions of this extract. *p*-CA is known to have many biological effects [12,13], but its activity is sometimes reported in a contradictory way. Therefore, we tested its cytotoxicity. Two cell lines, MCF7 and PC3, were selected, based on cytotoxic activity that was exerted by the extract from which *p*-CA was isolated, as reported in our previous works [10,11]. The mechanism of action of phenolic acid, which causes apoptosis, was previously reported for some cell lines and has been associated with the production of intracellular reactive oxygen species (ROS) [14]. Reactive oxygen species have been reported to affect the genetic expression of cells, affecting mainly the mitogen-activated protein kinase (MAPK) pathway [15]. We evaluated the gene expression of PC3 cells that were treated with *p*-CA in a microarray assay, which has not been previously reported, thereby contributing to knowledge of the cytotoxic effect of this phenolic acid. The main goal of the present study was to explore alternative mechanisms of action of *p*-CA and provide the first detection of constituents of active fractions from the *Hechtia glomerata* CHCl_3_-MeOH (1:1) extract.

## 2. Results and Discussion

### 2.1. Plant Material Extraction

Two solvents and one solvent mixture of different polarities were employed in the consecutive extraction of *H. glomerata* plant material: hexane, CHCl_3_/MeOH (1:1), and distilled water. The same extraction technique was used, but different volumes and repetitions were applied in the procedure, selecting them as necessary. Hexane maceration was needed only for defatting, which was applied once to the vegetal material to avoid the extraction of more polar compounds by co-solvation. This resulted in 9.5 g of a greasy dark yellow extract with a yield of 0.48%, weighted on the total amount of macerated leaves. CHCl_3_/MeOH extraction was repeated until the exhaustion of pigments from the ground leaves (6×). Our objective was to extract the highest quantity secondary metabolites. Therefore, a 1:1 mixture of CHCl_3_ and MeOH was used to widen the range of extracted molecules. This dark green extract had a dry weight of 87 g, representing a yield of 4.38% *w*/*w*. Finally, water extraction served to extract all polar compounds that were not obtained in the previous extraction, such as glycosylated and polyhydroxylated metabolites. A 21.01 g brown extract was obtained, with a yield of 10.51%, weighted on the sample of plant material that was used in the extraction.

The efficiency of the extractions was somewhat low in all macerations, with the water extract having the highest extraction yield. This was likely because of the high fiber content of the plant, which gave the material an extremely high volume despite its low weight and made it more difficult to reach intimate contact with the solvents during stirring.

### 2.2. Cytotoxic Activity of Extracts

The hexane extract had interesting activity against MCF7 cells (half maximal inhibitory concentration [IC_50_] = 27 ± 2.5 µg/mL) and low activity against PC3 cells (IC_50_ = 76 ± 5 µg/mL). The CHCl_3_/MeOH extract had significant cytotoxicity against PC3 cells (IC_50_ = 25 ± 4 µg/mL) and MCF7 cells (IC_50_ = 32 ± 2 µg/mL). These extracts had IC_50_ values that were lower than, or close to, the threshold of activity that is set by American National Cancer Institute guidelines (IC_50_ = 30 µg/mL) [16,17]. Thus, they were considered to have a medium to high cytotoxic effect on the evaluated cell lines. The aqueous extract was devoid of activity (Table 1).

The CHCl_3_/MeOH extract had cytotoxic effects on both PC3 and MCF7 cells. Therefore, we further investigated it because of its broad composition, which might reveal interesting compounds. The extract was fractionated by column chromatography (CC) [10], and its pooled fractions were tested to identify compounds that were responsible for the observed activities.

### 2.3. Cytotoxic Screening of Pooled Fractions

The results in Table 2 show the IC_50_ values for pooled fractions of the CHCl_3_/MeOH extract against both cancer cell lines, MCF7 and PC3. Pooled fraction **4** was the most active against both prostate and breast cancer cell lines. Therefore, we sought to determine its chemical composition to identify the active cytotoxic constituents.

### 2.4. HPLC-QTOF-MS Analysis of Cytotoxic Fraction **4**

Fraction **4** was analyzed using an Agilent instrument that was equipped with a biphenyl HPLC column and H_2_O/MeOH gradient as the mobile phase. This HPLC device was coupled to QTOF-MS. Each separated compound had its own mass spectrum, which was analyzed using the databases that are shown in Table 3. Table 3 shows the constituents of fraction **4**, together comprising a total of 22 compounds: 11 hydroxycinnamic acid (HCA) derivatives (62%), 10 flavonoids (11%), two coumarins (7%), two benzoic acid derivatives (2%), one pigment (10%), one lignan (3%), one fatty acid (2%), and five unknown compounds (10%). This fraction was analyzed using both positive and negative modes of detection.

Caffeic acid (CA), 2-*O*-*p*-coumaroyl glycerol, *p*-CA, 1-*O*-*p*-coumaroyl glycerol, dicaffeoyl glycerol, caffeoyl coumaroyl glycerol, and 3-(4-hydroxyphenyl-3-methoxy)propyl coumarate were identified in fraction **4**. These phytocompounds could be directly involved in the observed cytotoxic activity against both cell lines. In fact, plants of the genus *Tillandsia* and *Ananas* (Bromeliaceae) have already been shown to have the ability to synthesize these types of glyceryl HCA derivatives (e.g., *2*-*O*-*p*-coumaroyl glycerol, *1*-*O*-*p*-coumaroyl glycerol, and caffeoyl coumaroyl glycerol), which are known to have various biological activities [21,22]. The effects of the extracts of *Tillandsia recurvata* were shown to have potent effects on PC3 cells (IC_50_ = 2.4 µg/mL), and this activity was linked with the potent inhibition of protein kinases (K_d_ = 8-14 µg/mL) that are implicated in cancer growth and proliferation [23]. *p*-CA and its derivatives are also known to affect the viability of numerous tumor cell lines, such as MCF7, with some derivatives having IC_50_ values as low as 6.9 µM [12], but free *p*-CA is usually less active, with an IC_50_ > 500 µM [13]. The cytotoxic activity of CA has also been reported against PC3 cells, with an IC_50_ of 177.62 µM [24]. Compared with the observed activity of paclitaxel (39.5 ± 4.6 and 15.5 ± 3.5 nM for MCF7 and PC3 cells, respectively), the reported activities of phenolic acids are significantly lower. All of the identified compounds whose cytotoxicity has already been reported are listed in Table 4.

In some cases, these compounds are known to produce ROS in the culture medium that is used for cell lines and microorganisms, such as H_2_O_2_, which could explain this fraction’s cytotoxicity [25]. 3-(4-Hydroxyphenyl)propyl coumarate and 3-(4-hydroxyphenyl-3-methoxy)propyl coumarate could also be cytotoxic agents, in which they have a similar structure to caffeic acid phenethyl ester (CAPE), which is reported to have anticancer activity through inhibition of the phosphoinositide 3 kinase/protein kinase B (PI3K/Akt) signaling pathway and inhibition of the transcription factor nuclear factor κB (NF-κB), thereby lowering cell survival [26,27]. The reported activity for these compounds could explain the cytotoxicity that was observed in fraction **4**, but further studies of pure compounds are necessary to confirm this possibility.

### 2.5. Cytotoxicity of p-CA

*p*-CA was previously isolated and purified through several column chromatography separations of the CHCl_3_/MeOH (1:1) extract. The pure compound was then structurally characterized by ^1^H and ^13^C nuclear magnetic resonance [10]. Furthermore, *p*-CA was identified as one constituent of the cytotoxic fraction, with an estimated 6.05% of the total weight. Although at a lower concentration, phenolic acid was also found in fraction **3**, but its lack of activity suggested that additional compounds are involved in the observed effect. Thus, we tested this compound for cytotoxic activity against MCF7 and PC3 cell lines.

The results showed that *p*-CA had weak cytotoxic effects against PC3 cells (IC_50_ = 1.1 ± 0.2 mM). The compounds that were contained in the active fraction likely act synergistically. Nevertheless, cellular assays are needed to evaluate and prove the source of the observed effects. *p*-CA was devoid of activity against MCF7 cells (IC_50_ > 1.2 mM), meaning that among the other identified compounds, one or several other active compounds should act synergistically against MCF7 cells. The IC_50_ values of the positive control paclitaxel were 39.5 ± 4.6 and 15.5 ± 3.5 nM against MCF7 and PC3 cells, respectively.

Compared with other polyphenolic and antioxidant compounds that were reported to affect the growth of cancer cells, such as quercetin and kaempferol [30,31], *p*-CA activity was weaker. In fact, quercetin was reported to have an IC_50_ > 120 µM against PC3 cells with 24 h treatment [32], but this concentration greatly decreased when the time of exposure increased, reaching values as low as 36 ± 1.98 µM. Similar results were also observed for MCF7 cells, with IC_50_ values of a comparable magnitude. Quercetin and kaempferol exert cytotoxic effects and protect against inflammation by directly inhibiting various MAPKs, such as AKT and MAPK/extracellular signal-regulated kinase (ERK) kinase 1 (MEK1). Similar results were also obtained with resveratrol, another polyphenolic compound that is also able to prevent neoplastic transformation of cells by affecting the same pathways as previously cited for flavonoids [33]. Another more structurally related plant metabolite that has a similar cytotoxic mechanism of action is rosmarinic acid, which also inhibits protein kinases [34], such as MARK4, which regulates the early cell division step and causes apoptosis. These polyphenols directly inhibit MAPKs. *p*-CA could also act in the same way but with lower potency. Therefore, because the PC3 cell line was weakly sensitive to *p*-CA, we determined the cytotoxic mode of action of this molecule using a microarray assay.

### 2.6. Microarray Assay

As described in the Material and Methods section, two cultures of PC3 cells were prepared: one that was treated with 0.9 mM (IC_30_) of *p*-CA and a control that was only exposed to dimethylsulfoxide (DMSO; 0.6%). We decided to use a sublethal concentration of the compound to induce responses from cancer cells outside the apoptotic process. Afterward, total RNA that was extracted from each culture was utilized for the synthesis of complementary DNA (cDNA), which was then used in the microarray analysis. From the latter, 230 genes were upregulated, and 177 were downregulated. Theses differentially expressed genes were then submitted to the Database for Annotation, Visualization and Integrated Discovery (DAVID) bioinformatics platform for annotation. The functional categories for each annotated gene are shown in Table 5. Of the reported biological functions, among the upregulated genes, the most prominent were effects on phosphoproteins (45.65%) and the nucleus (36.09%). Phosphoprotein was also observed as the highest category for downregulated genes (55.37%), along with membrane effects (45.76%).

The up- and downregulated genes underwent a bioinformatic analysis using DAVID software. The bioinformatic study showed that the process of protein phosphorylation was mainly affected by *p*-CA treatment. Many of the dysregulated genes were shared by different signaling pathways, especially MAPK, adenosine monophosphate-activated protein kinase (AMPK), Ras, PI3K/AKT, and cancer pathways. However, the effects on these classes of proteins were seen for both the up- and downregulated gene sets, which contrast with the reported mode of action for ROS, which activates these proteins to cause apoptosis [15]. Therefore, *p*-CA may also directly interact with some of the protein kinases, causing their inhibition and subsequent upregulation of the affected enzymes.

These upregulated proteins include MAPK8, MAPK-interacting serine/threonine-protein kinase 2 (MKNK2), MAPK kinase kinase 1 (MAP3K1), and Ser/Thr protein kinase 3 (STK3), which are all MAPKs that share a common phosphorylation site. The downregulated genes that encode MAPK proteins include AKT2, STK11, and neurotrophic receptor tyrosine kinase 3 (NTRK3), which can be activated by ROS to induce the apoptotic process. AKT2 is an isoform of three AKT proteins (AKT1, AKT2, and AKT3) and plays a central role in the AKT/PI3K signaling pathway, which is activated by oxidative stress that is caused by ROS. The same applies to NTRK3 and STK11, which are involved in either the AMPK pathway or central carbon metabolism in cancer and are also activated by oxidative stressors, such as ROS [35,36]. The activation of STK11 could also be involved in the activation of phosphofructokinase 1 (PFK1) and PFK2, which are part of the glycolysis process [37]. These proteins and the AMPK pathway are of particular interest because of their involvement in cancer multidrug resistance and their use for the prediction of clinical outcome of major cancer types [38].

MAPK8, also known as c-Jun N-terminal kinase 1 (JNK1), is a protein that is ubiquitously expressed in all mammalian tissues and involved in many signaling pathways. It is implicated in the development of many cancer types, such as skin cancer and hepatocellular carcinoma. It is a part of JNK isomers (JNK1, JNK2, and JNK3), and it is well known to modulate various cellular functions, including cell survival, proliferation, death, and differentiation. Its activation in cancer leads to negative regulation of the tumor suppressor p53, but no important role of MAPK8 has been reported in prostate cancer. Additional proof of the importance of JNK proteins came from a large-scale sequencing analysis of various tumor types that identified somatic mutations that linked these proteins to the development of cancer [39]. Because of its pivotal function in cell proliferation and apoptosis, this protein has already been targeted for development of the inhibitors JNKI1 and BI-78D3 [40,41]. The latter interfered with kinase binding to the JNK-interacting protein 1 (JIP1) scaffold, promoting the programmed cell death of cancer cells. The activity of these inhibitors was shown to be effective in both human cells and an in vivo animal model, but they were not clinically tested [42]. Another example of a MAPK8 inhibitor is the JNK inhibitor AS602801, the ion synergy of which with enzalutamide killed prostate cancer cells in vitro [43]. Furthermore, polyphenols (e.g., the flavonoid quercetagetin) are reported to act as natural JNK1 inhibitors [44].

Among other genes that were found to be desregulated by *p*-CA treatment were the MDM2 proto-oncogene (upregulated) and MDM4 (downregulated), which are critical regulators of the tumor suppressor p53. These proteins were reported by Qin et al. to be targets for prostate cancer therapy that inhibited their proliferation and invasion in a p53- and androgen-independent manner. They are involved in such processes as initiation, progression, metastasis, and chemotherapeutic resistance [45]. The present results showed the inhibition of MDM2 by either ROS or *p*-CA, additionally reported was the downregulation of MDM4, which is involved in the p53 response to genotoxic stress [46]. Eukaryotic translation initiation factor 4H (EIF4H) was also found to be downregulated by *p*-CA treatment. This protein is involved in cancer cell invasion and metastasis and usually upregulated in adenocarcinomas [47]. Moreover, the desregulation of translation initiation factors is associated with carcinogenesis. In some cases, such as lung adenocarcinoma, the depletion of EIF4H enhances cancer sensitivity to chemotherapy [48]. Finally, inhibitor of NF-κB kinase subunit γ (IKBKG), which is an essential modulator of NF-κB, was also upregulated. The activation of this transcription factor is usually associated with oxidative stress, which is consistent with the effects of ROS that are produced by *p*-CA treatment [49].

*p*-CA is a dietary hydroxycinnamic acid with antioxidant, antimicrobial, antiviral, antimutagenesis, anticancer, analgesic, antipyretic, hypopigmenting, antiulcer, antiarthritis, antiplatelet aggregation, and anxiolytic activities. *p*-CA was also shown to mitigate diabetes [12]. These types of MAPK inhibitors have been reported to enhance the antitumor effects of cytotoxic drugs [50]. These properties make *p*-CA a potential candidate for the development of new anticancer drugs and chemotherapeutic-enhancing formulations that can re-sensitize therapeutic-resistant cancer types.

### 2.7. Molecular Docking and Molecular Dynamics Simulations

To evaluate the possible binding modes of *p*-CA on selected Ser/Thr protein kinases, a three-dimensional model of the proteins was retrieved from the Protein Data Bank (PDB). For MAPK8, the structure (PDB ID: 2G01; resolution: 3.50 Å) was published by Liu et al. as the crystallized structure of the enzyme with a novel pyrazoloquinolone inhibitor (JNK inhibitor II) [51]. The pdb file contained the protein in a dimeric complex. Therefore, it was processed to leave only the monomeric structure (chain A). Different authors recently reported using UCSF Chimera program in molecular modeling studies which included ligand-protein and protein-protein molecular docking experiments [52,53,54,55]. To identify the binding site of *p*-CA in the selected proteins, a molecular docking study was performed. The blind docking results showed that phenolic acid bound in a cavity that was the main catalytic site of MAPK proteins that were employed in the simulations (Figure 1a). All of these proteins, although they have different amino acid sequences, shared the same tertiary structure of the active site. Some inhibitors bound to this region, such as JNK inhibitor II (Figure 1b) and Xmu-MP-1. The latter is a cytotoxic benzodiazepine that binds STK3 (Figure 1c). *p*-CA binding to the same active site as the inhibitors suggests similar activity. This binding site was then selected because it gave the best interaction with the ligand and because of its obvious therapeutic implications. The best pose of *p*-CA was selected based on the AutoDock-Vina scoring function (−6.5 kcal/mol) and was employed in the molecular dynamics simulations.

The results of the simulations showed dynamic interactions of the hydroxycinnamic acid with amino acids of the active site of the selected proteins (Figure 2). The longest and strongest interactions were observed between *p*-CA and the MAPK8 active site, reporting salt bridges and H-bonds that were mediated by water molecules between the carboxylic group of *p*-CA and residues Tryptophane 7 (Trp7), Methionine 130 (Met130), and Aspartate 131 (Asp131). The latter lasted between 43% and 48% of the simulation time, making them medium strength. The longest and thus strongest interactions were H-bonds between the carbonyl group and Asparagine 133 (Asn133, 76%) and between the phenol moiety and Glutamate 128 (Glu128, 92%). Furthermore, a negatively charged interaction, together with a hydrophobic and polar one, was visible between *p*-CA and the protein backbone, which further increased the strength of this interaction.

From the molecular dynamics simulation, we also approximated variation of the binding free energy during the interaction between *p*-CA and MAPK8 (∆E_bound_ = −1.927 kcal/mol) using the linear interaction energy (LIE) equation. The reported negative value of free energy denotes spontaneity of the binding process between the phenolic acid and protein. However, the goodness of this prediction needs to be validated by inhibition of the protein in vitro.

As a future goal, the active fractions will be further fractionated to simplify their composition and proceed to the isolation and structural characterization of their active principles. We will also prepare semi-synthetic derivatives of some of the identified active compounds, such as CAPE-like molecules, to confirm their presence in active fractions and perform a structure-activity relationship study.

## 3. Materials and Methods

### 3.1. Plant Material

The plant vegetal material was collected in Rayones (25°01′00”N 100°05′00”W, 906 m above sea level), Nuevo León, Mexico. *H. glomerata* collection and identification were realized by the biologist Mauricio González Ferrara on February 15, 2017. A voucher specimen (reference no. 028029) was deposited at the herbarium of the Faculty of Biology, Universidad Autónoma de Nuevo León. The plant name can be found at http://www.theplantlist.org/ [10].

### 3.2. Extract Preparation

Plant leaves (10.5 kg) were cut into small pieces and dried in the dark. The dried and ground material (2.29 kg) was macerated at room temperature sequentially with hexane (Hex; 10 L × 1; 24 h), CHCl_3_/MeOH (1:1 mixture; 10 L × 6; 24 h), and distilled water (2.5 L × 1; 24 h). The organic extracts were dried in vacuo (<40 °C), whereas the aqueous extract was lyophilized. The above procedure resulted in Hex (9.5 g; 0.48%), CHCl_3_/MeOH (87 g; 4.38%), and aqueous (21.01 g; 10.51%) extracts. These were kept at −20 °C until further use [10]. Organic solvents used were 95% pure (Baker Company, Sanford, ME, USA).

### 3.3. Fractionation of CHCl_3_/MeOH Extract by Column Chromatography

The CHCl_3_/MeOH extract (87 g; 4.38%) was fractionated by passing through CC on 2 kg of silica gel with 230–400 mesh (EMD Chemicals, Burlington, MA, USA) and eluted with a gradient of Hex/EtOAc/MeOH. The composition of the mobile phase was gradually increased in polarity based on the continuous thin layer chromatography (TLC) analysis of the produced fractions. The gradient started at 100% hexane, and the polarity of the eluent was increased by 10% for every two dead volumes (Vo) by adding EtOAc until 100% was reached. Afterward, the same procedure was repeated, but MeOH was added as a more polar solvent to the eluent until this solvent reached 100%. Each fraction had a volume of 500 mL and was pooled into a total of eight fractions. The TLC analysis of the fractions and pooled fractions was performed on Silica gel 60 F254 precoated on aluminum (Merck, Darmstadt, Germany) or C18 TLCs precoated on glass. The TLCs were visualized under ultraviolet light (254 and 365 nm) and stained with Ce(SO_4_)_2_/H_2_SO_4_ solution [10].

### 3.4. Cell Lines and Cytotoxicity Assay

The cell lines were purchased from the American Type Culture Collection (ATCC): prostate adenocarcinoma (PC3, ATCC CRL-1435) and breast adenocarcinoma (MCF7, ATCC HTB-22). MCF7 cells were maintained in Dulbecco’s modified essential medium (DMEM; Invitrogen, Thermo Fisher Scientific, Waltham, MA, USA). PC3 cells were grown in RPMI 1640 medium (Invitrogen, Thermo Fisher Scientific, Waltham, MA, USA). Both media were supplemented with 10% fetal bovine serum (FBS), 2 mM L-glutamine, 40 µg/mL gentamicin, and penicillin-streptomycin (100 U/mL penicillin and 100 µg/mL streptomycin). The cultures were incubated at 37 °C in a humidified atmosphere of 5% CO_2_. These cell lines were tested for cytotoxic activity of the extracts, fractions, and pure compounds. Crude extracts, fractions, and pure compounds were dissolved in DMSO (≤ 0.6%) according to the appropriate supplemented culture medium. To determine the cytotoxic effects of the extracts, 5000 cells/well were seeded in a 96-well cell culture plate and treated with different concentrations of extracts, fractions, and pure compounds. The cells were exposed to the extracts, fractions, and compounds for 72 h at 37 °C with 5% CO_2_. The number of viable cells was determined using the CellTiter 96 AQueous One Solution Cell Proliferation Assay kit (Promega, Madison, WI, USA) or WST-1 reagent (Sigma Aldrich, St. Luis, MO, USA) according to the manufacturer’s instructions. Cell viability was determined by reading absorbance at 450 nm using an automated enzyme-linked immunosorbent assay reader. The experiments were conducted in triplicate in three independent tests. The data were analyzed using Prism 7.0 software, and IC_50_ values were determined by regression analysis [56].

### 3.5. HPLC-QTOF-MS Analysis

Fraction **4** was analyzed using an Agilent Technologies series 1200 instrument with an electrospray ionization (Agilent JetStream, Santa Clara, CA, USA) ion source and QTOF Agilent technologies 6530A model detector under the following conditions. For the stock solution, 15 mg of each fraction was dissolved in 10 mL of CH_3_CN/acetone/DMSO (20:70:10). Properly diluted samples were separated into their components by passing through a Phenomenex Kinetex Biphenyl 50 × 2.1 mm, 2.6 μm HPLC column. The components of the fractions were separated with a gradient of water and methanol (both with 0.1% formic acid) with a flux of 0.4 mL/min. The gradient began with 10% methanol, reaching 100% in 8 min. It was kept at this percentage for 1.5 min, and then the gradient went back to 10% methanol in 0.1 min. The ion source was electrospray. The detector was set in both the positive and negative modes, with a range of acquisition between 40 and 2000 Da. The difference in potential at the source was 4000 V. The octapole frequency was 750 V. The fragmentor voltage was 150 V. The separated components were identified primarily using the built-in METLIN, MassBank, and In-house databases. Alternatively, the molecules were tentatively identified using the FooDB, ReSpect for Phytochemicals (ReSpect, RIKEN Plant Science Center, Yokohama, Japan), PhytoHub and Plant Metabolic Network (PMN) databases or manually, and comparisons were made with the literature. The complete procedure of bioassay-guided fractionation is shown in Figure 3.

### 3.6. Cell Treatment and Isolation of Total RNA

PC3 cells were treated in a six-well plate with 10^6^ cells per well for 24 h with *p*-CA at its IC_30_ (0.9 mM) and 0.6% DMSO or only with DMSO at the same concentration. After incubation, total RNA was extracted from both cultures that were treated with either the compound or vehicle using TRIzol Reagent (Invitrogen, Carlsbad, CA, USA) according to the manufacturer’s instructions. The RNA quality was assessed by agarose gel (1%), and total RNA was quantified in a NanoDrop 2000 Spectrophotometer (Thermo Fisher Scientific, Waltham, MA, USA).

### 3.7. Microarray Chip Printing

Microarray hybridization was performed on a human 35K chip, surveying a total of 35,764 genes (whole genome). This chip was printed as the following. The human 70-mer oligo library from OPERON Oligo Sets (http://omad.operon.com/) was resuspended to 40 µM in Micro Spotting solution (Telechem International, Los Altos, CA, USA). A SuperAmine coated slide (25 × 75 mm; TeleChem International) was printed in a single copy and fixed at 80 °C for 4 h. For pre-hybridization, the slide was rehydrated with water vapor at 60 °C and fixed with two cycles of ultraviolet light (1200 J). After boiling for 2 min at 92 °C, the slide was washed with 95% ethanol for 1 min and prehybridized in 5× saline-sodium citrate (SSC) buffer, 0.1% sodium dodecyl sulfate (SDS), and 1% bovine serum albumin for 1 h at 42 °C. The slide was then washed and dried for further hybridization.

### 3.8. Labeled cDNA Synthesis and Microarray Assay

RNA quantity was adjusted to 10 µg for both treated and untreated controls. The modified cDNA that possessed the nucleotide Alexa Fluor-uridine was synthesized using the First-Strand cDNA labeling kit (Invitrogen, Life Technologies, Grand Island, NY, USA). The fluorescent dyes Alexa555 and Alexa647 were used to label cDNA, and their incorporation was analyzed by reading absorbance at 555 and 650 nm, respectively.

For hybridization, equal quantities of labeled cDNA for treated and untreated samples were added to the chip and hybridized using UniHyb hybridization solution (TeleChem International). The arrays were incubated for 14 h at 42 °C and then washed three times with 1× SCC and 0.05% SDS at room temperature. The acquisition and quantification of array images were performed in GenePix 4100A with its accompanying GenePix software (Molecular Devices, San José, CA, USA). All images were captured using 10 µm resolution. For each spot, the Alexa555 and Alexa647 density mean value and background mean value were calculated using ArrayPro Analyzer software (Media Cybernetics, Rockville, MD, USA).

### 3.9. Microarray Data Analysis

The data were statistically analyzed using free GenArise software that was developed in the Computing Unit of Cellular Physiology Institute at Universidad National Autónoma de México (UNAM) (http://www.ifc.unam.mx/genarise/). A number of transformations were performed by GenArise, including background correction, lowness normalization, intensity filter, replicate analysis, and selecting differentially expressed genes. The goal of this software is to identify genes with good evidence of being differentially expressed by calculating an intensity-dependent z-score. This is performed using a sliding window algorithm that calculates the mean and standard deviation within a window that surrounds each data point. The algorithm then defines a z-score, where z measures the number of standard deviations a data point is from the mean using Equation (1) for z-score determination.
(1)$$z_{i}=\frac{\left(R_{i}-\mathrm{mean}\left(R\right)\right)}{\mathrm{SD}(R)}$$
where z_i_ is the z-score for each element, R_i_ is the log-ratio for each element, and SD(R) is the standard deviation of the log-ratio. Based on this criterion, elements with a z-score > 2 standard deviations would be significantly differentially expressed genes, which were then selected for the gene expression analysis [57,58]. The complete lists of selected up- and downregulated genes are reported in the Appendix A. Both lists of genes were analyzed separately for biological pathway identification using bioinformatics tools, such as DAVID 6.8 [59], Kyoto Encyclopedia of Genes and Genomes (KEGG) [60], and the National Center for Biotechnology Information (NCBI) [61].

### 3.10. Molecular Docking

The crystalline structure of the selected proteins was retrieved from PDB [62]. Incomplete protein structures and sequence gaps were filled through the generation of theoretical models using online platforms, such as I-TASSER, Galaxy WEB, and SWISS-MODEL [63,64,65]. The three-dimensional model of *p*-CA (ligand) was drawn using the molecular builder Avogadro 1.2.0 [66]. These files were then prepared for docking using the open-source molecular modeling program Chimera (University of California, San Francisco) [67] and Docking Preparation wizard (DockPrep-AutoDock Tools). Small protein chain gaps or missing residues were replaced using the Dunbrack 2010 rotamers library [68]. The AutoDock-Vina version 1.1.2 plug-in was used in Chimera for the docking experiments. The Opal web service was used as the host for the docking simulations (server: http://nbcr-222.ucsd.edu/opal2/services/vina_1.1.2). The whole protein surface was defined as the search space, with a grid size in the range of 60-80 Å^3^, adjusting the volume manually so that each protein would be at its center. Default optimization parameters for the docking and pose prediction analysis were used in the simulation. The best pose was selected based on the molecular docking score and interactions between the ligand and protein side chains, favoring the lowest score with the highest number of interactions.

### 3.11. Molecular Dynamics Simulations

Molecular dynamics simulations were performed using the Schrödinger release 2019-3 programs Desmond Molecular Dynamics System (D. E. Shaw Research, New York, NY, USA, 2019) and Maestro-Desmond Interoperability Tools (Schrödinger, New York, NY, USA, 2019). The ligand-protein complex, containing the best pose of *p*-CA in the enzyme active site, was enclosed in the orthorhombic boundary box and solvated by adding water molecules in the SPC model. Physiological-like conditions were applied to the system, such as salinity (0.15 M), temperature (310.15 K), and pressure (1.013225 bars). To maintain these conditions, Nose-Hoover thermostat [69,70] and Martyna-Tobias-Klein barostat [71] algorithms were used. The length of the simulation was set to 10 ns, and the recording interval for the trajectory was every 10 ps (1000 frames) [72].

### 3.12. Binding Energy Approximation

The resulting simulation frames were clustered by obtaining the representative structure for each ligand-protein complex that was analyzed. From these structures, the electrostatic and van der Waals energies for protein-bound and free ligand were calculated. These parameters were then used in the approximation of binding energy (∆E_bound_) of *p*-CA to the tested proteins by employing the LIE equation (2):(2)$$∆E_{\mathrm{bound}}=\alpha \left(E_{\mathrm{lp}}^{\mathrm{vdW}}-E_{\mathrm{ls}}^{\mathrm{vdW}}\right)+\beta \left(E_{\mathrm{lp}}^{\mathrm{el}}-E_{\mathrm{ls}}^{\mathrm{el}}\right)+\gamma$$
where $E_{\mathrm{lp}}^{\mathrm{el}}$, $E_{\mathrm{ls}}^{\mathrm{el}}$, $E_{\mathrm{lp}}^{\mathrm{vdW}}$ and $E_{\mathrm{ls}}^{\mathrm{vdW}}$ are the ligand-protein and ligand-solvent (free) electrostatic and van der Waals potential energies from the molecular dynamics simulation. α, β, and γ were scaling factors that were used with default values of 0.16, 0.5, and 0 [73].

## 4. Conclusions

In conclusion, bioassay-guided fractionation of the CHCl_3_/MeOH extract of *H. glomerata* Zucc., and the HPLC-QTOF-MS analysis of its active fractions complemented the data that were reported by previous studies of the extracts of this plant. The identified compounds, specifically phenolic acids, were suitable for use as platforms for the development of new anticancer drugs. Furthermore, the presence of these compounds is useful for generating additional hypotheses of the properties of *H. glomerata* and its ethnomedical use. The overexpression of MAPK proteins that was observed in the microarray assay and resulting interactions between *p*-CA and these proteins, revealed by the molecular docking and molecular dynamics simulations, provide evidence of the possible inhibition of these kinases, such as MAPK8, by the phenolic acid. Moreover, the effects of *p*-CA treatment on the expression of genes that are involved in the cell adhesion and apoptosis modulation of PC3 cells and already known health benefits of *p*-CA consumption make it a candidate for the development of new anticancer drugs or cytotoxicity-enhancing formulations.

## Figures and Tables

**Figure 1 molecules-26-01096-f001:**
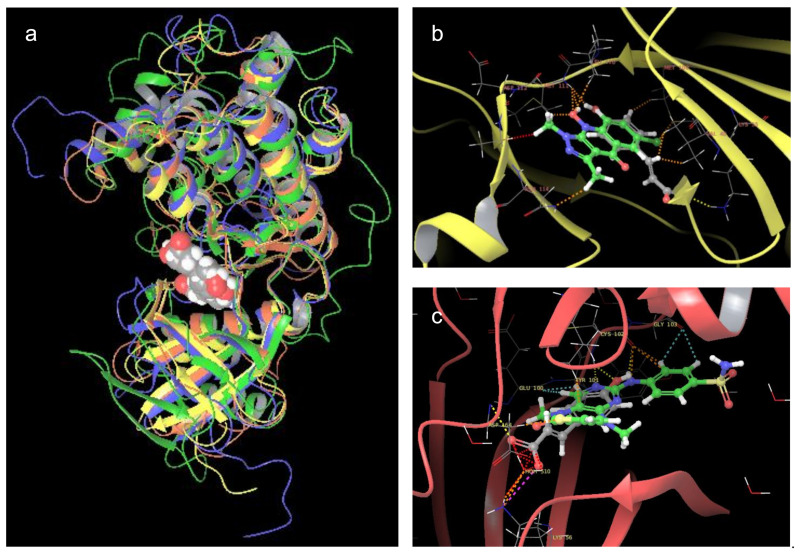
(**a**) Conserved protein structure motif alignment for mitogen activated protein kinase kinase kinase 1 (MAP3K1, green), mitogen activated protein kinase 8 (MAPK8, yellow), MAP kinase-interacting Serine/Threonine kinase 2 (MKNK2, blue), and human Serine/Threonine-protein kinase 3 (STK3, red) that bound with the best poses of *p*-coumaric acid (*p*-CA) for each protein. (**b**) JNK inhibitor II (green) and *p*-CA bound to MAPK8. (**c**) Xmu-MP-1 (green) and *p*-CA bound to STK3. These images were made with the Maestro suite.

**Figure 2 molecules-26-01096-f002:**
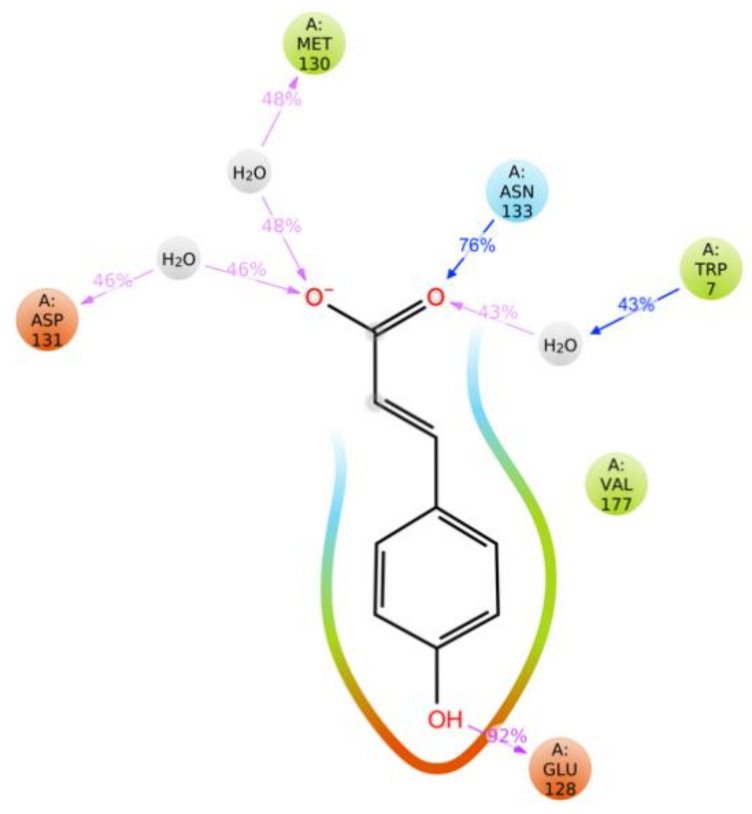
Dynamic interaction of *p*-CA with amino acids of the MAPK8 active site. Blue arrows indicate the direction of an H-bond donation. Pink arrows indicate H-bond acceptance or salt bridges. The percentages represent the strength of the interaction relative to simulation time. Higher percentages indicate longer interactions during the simulation. This image was obtained from the molecular dynamics simulation report that was produced by Desmond software.

**Figure 3 molecules-26-01096-f003:**
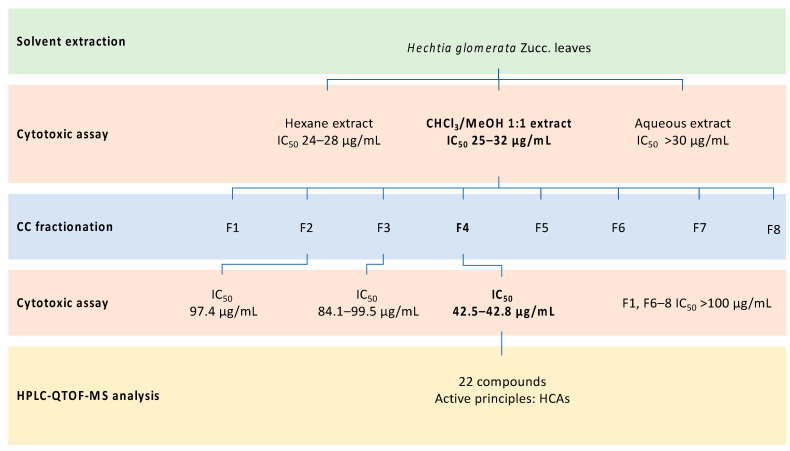
Bioassay-guided fractionation procedure: exhaustive solvent extraction of vegetal material; cytotoxic assays of obtained extracts; column chromatography (CC) fractionation of active extract; cytotoxic assays of fractions against cancer cell lines affected by the whole extract; high-performance liquid chromatography-quadrupole/time-of-flight-mass spectrometry (HPLC-QTOF-MS) analysis of active fraction.

**Table 1 molecules-26-01096-t001:** Cytotoxic activity of *H. glomerata* extracts.

Extract	IC_50_ ^a^ ± SD ^b^ (µg/mL)	
	PC3 ^c^	MCF7 ^d^
Hexane	76 ± 5	27 ± 2.5
C/M ^e^	25 ± 4	32 ± 2
Aqueous	81 ± 3	71 ± 3
**Positive control**	**IC_50_ ± SD (µg/mL)**	
Paclitaxel	13.2 ± 3 × 10^−3^	33.7 ± 3.9 × 10^−3^

^a^ IC_50_: half maximal inhibitory concentration; ^b^ SD: standard deviation; ^c^ PC3: prostate adenocarcinoma cells; ^d^ MCF7: breast adenocarcinoma cells. ^e^ CHCl_3_/MeOH (1:1).

**Table 2 molecules-26-01096-t002:** Cytotoxic activity of pooled fractions of the CHCl_3_/MeOH extract.

Pooled Fraction (g)	IC_50_ (µg/mL)	
	MCF7	PC3
1 (1.07)	>100	>100
2 (11.09)	>100	97.4 ± 14.8
3 (18.30)	99.5 ± 7.3	84.1 ± 8.4
4 (9.60)	42.5 ± 8.6	42.8 ± 7.2
5 (6.78)	>100	>100
6 (4.98)	>100	>100
7 (7.38)	>100	>100
8 (12.70)	>100	>100
**Positive control**	**IC_50_ (µg/mL)**	
Paclitaxel	33.7 ± 3.9 × 10^−3^	13.2 ± 3 × 10^−3^

**Table 3 molecules-26-01096-t003:** High-performance liquid chromatography-quadrupole/time-of-flight-mass spectrometry (HPLC-QTOF-MS) analysis of cytotoxic pooled fraction **4**.

Retention Time (min)	Compound	Calculated Formula	[M − H]^−^	[M + H]^+^	Fragments	Area (%)	Ref.
1.001	Protocatechuic acid	C_7_H_6_O_4_	153.0189	-	[M − H]^−^: 109	0.89	[18]
1.732	Salicylic acid	C_7_H_6_O_3_	137.0239	-	[M − H]^−^: 93	0.86	^a, b^
3.129	Caffeic acid	C_9_H_8_O_4_	179.0342	-	[M − H]^−^: 135	0.24	^a^
4.416	2-*O*-*p*-Coumaroyl glycerol	C_12_H_14_O_5_	237.0768	239.0900	[M − H]^−^: 119, 145, 163 [M + H]^+^: 147, 119	7.22	[18]
4.737	*p*-Coumaric acid	C_9_H_8_O_3_	163.0400	-	[M − H]^−^: 119	6.05	[19]
4.99	1-*O*-*p*-Coumaroyl glycerol	C_12_H_14_O_5_	237.0767	-	[M − H]^−^: 119, 145, 163	12.18	^c^
5.384	Unknown	NA	-	275.1300	[M + H]^+^: 79, 197, 179, 135	3.84	^m^
6.561	Dicaffeoyl glycerol isomer	C_21_H_20_O_9_	-	417.1200	[M + H]^+^: 163, 145, 237	3.50	^c^
6.741	Dicaffeoyl glycerol	C_21_H_20_O_9_	415.1044	417.1200	[M − H]^−^: 253, 161, 179, 135, 237 [M + H]^+^: 163, 145, 164, 237	14.53	[20]
7.063	Caffeoyl coumaroyl glycerol	C_21_H_20_O_8_	399.1102	401.1200	[M − H]^−^: 163, 253, 119, 145, 135, 235, 179 [M + H]^+^: 147, 163, 145, 119	3.66	^m^
7.514	Naringenin derivative	C_24_H_20_O_8_	-	437.1200	[M + H]^+^: 437, 177, 251, 147, 243, 221	2.22	^d^
7.569	Unknown	C_17_H_14_O_7_	-	331.0800	[M + H]^+^: 331, 315, 298, 273, 281	3.83	^m^
7.837	Chlorophyll derivative	NA	-	843.2300	[M + H]^+^: 843, 555, 844, 597, 556	10.32	^d^
8.013	Unknown	C_19_H_18_O_9_	-	391.1000	[M + H]^+^: 391, 361, 389, 375, 358	2.70	^m^
8.014	Coumarin isomer	C_9_H_6_O_2_	-	147.0400	[M + H]^+^: 91, 119, 147, 65	1.84	^m^
8.168	Coumarin derivative	C_20_H_22_O_6_	-	359.1500	[M + H]^+^: 147, 119	5.32	^m^
8.215	3-(4-Hydroxyphenyl)propyl coumarate	C_18_H_18_O_4_	297.1151	-	[M − H]^−^: 119, 145, 163 [M + H]^+^: 107, 147, 135	4.33	^m^
8.297	3-(4-Hydroxyphenyl-3-methoxy) propyl coumarate	C_19_H_20_O_5_	327.1255	-	[M − H]^−^: 163, 119, 145, 177, 133, 312, 299	3.02	^m^
8.37	Kaempferol or cyanidin derivative	C_18_H_16_O_7_	-	345.1000	[M + H]^+^: 345, 312, 329, 330, 287	3.37	^d^
8.784	Palmitoleic acid	C_16_H_30_O_2_	-	255.2300	[M + H]^+^: 69, 67, 83, 55, 81, 97, 95, 71, 43, 135, 93, 121, 107, 97, 149, 67, 109	1.71	^m^
9.156	Demethylnobiletin	C_20_H_20_O_8_	-	389.1200	[M + H]^+^: 389, 356, 331, 373, 359, 374	5.74	^e^
9.453	Syringaresinol	C_21_H_22_O_9_	-	419.1300	[M + H]^+^: 419, 389, 371, 420, 390	2.63	^m^

^a^ METLIN; ^b^ MassBank; ^c^ In-house database; ^d^ ReSpect; ^e^ FooDB; ^m^ Manual.

**Table 4 molecules-26-01096-t004:** Active phenolic compounds identified in fraction **4**.

Compound	Biological Effects	IC_50_/MIC ^a^	Target/Cells	Source	Ref.
1-*O*-*p*-Coumaroyl glycerol	-	-	-	*Ananas cumosus*	[22,28]
2-*O*-*p*-Coumaroyl glycerol	Antimicrobial		MR*SA* ^b^		
Caffeic acid	Cytotoxicity	177.62 µM	PC3	Purple potato extract	[24]
Caffeoyl coumaroyl glycerol	Cytotoxic effects	-	-	*Tillandsia streptocarpa*	[29]
Dicaffeoyl glycerol		-	-		
*p*-Coumaric acid	ROS, anti-angiogenesis, AKT and ERK signaling pathways	>500 µM	MCF7, murine tumors, N2a, ECV304	-	[13]

^a^ MIC: minimal inhibitory concentration ^b^ MR*SA*: meticillin-resistant *Staphylococcus aureus*.

**Table 5 molecules-26-01096-t005:** Functional category of up- and downregulated genes from the microarray analysis performed with PC3 cells that were treated with *p*-CA.

Functional Category	Upregulated Genes (%)	Downregulated Genes (%)
Acetylation	-	23.73
Activator	9.13	-
Apoptosis	4.78	-
ATP-binding	10.87	-
Calcium	7.39	-
Cell membrane	-	22.60
Coiled coil	19.13	-
Developmental protein	8.70	-
Differentiation	6.52	-
Disulfide bond	-	23.16
DNA damage	3.48	-
DNA-binding	18.70	-
EGF-like domain	3.91	-
G-protein coupled receptor	-	10.17
Kinase	-	7.91
Lipoprotein	6.52	-
Magnesium	-	6.21
Membrane	-	45.76
Metal-binding	29.57	23.16
Nucleotide-binding	11.74	-
Nucleus	36.09	-
Phosphoprotein	45.65	55.37
Proto-oncogene	-	3.95
Receptor	-	15.25
Repressor	5.22	-
Sugar transport	-	1.69
Transcription regulation	22.17	-
Transducer	-	10.17
Transferase	-	14.12
Transmembrane	-	36.16
Transport	-	14.12
Ubl conjugation	-	14.69
Zinc	21.74	-
Zinc-finger	18.70	-

## Data Availability

The complete microarray results were published on the Gene Expression Omnibus (GEO) database (NCBI) and are available under the Series accession number GSE155668.

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
