# Peer review of "Cytotoxic Fractions from Hechtia glomerata Extracts and p-Coumaric Acid as MAPK Inhibitors"

_molecules, 2021, doi:10.3390/molecules26041096_

Round 1

Reviewer 1 Report

The manuscript by Stefani and colleagues extends their analysis of bioactivity of Hechtia glomerata extracts to the investigation of cytotoxicity. The introduction is a bit convoluted and somewhat unclear in terms of existing data. Since the analysis of the extract contents has already been published, this should be completely clarified in the manuscript. In addition, more details on identification of the extract compounds should be provided. Furthermore, the cytotoxicity is very low compared to the standard, which reduces interest. Interestingly, individual fractions had lower toxicity than the full extract, suggesting that there is not a single agent at work. The low toxicity of p-coumaric acid definitely reduces enthusiasm for this work as does the prior studies of this compound. ROS are mentioned as important, and yet no studies on ROS were performed. This should be clarified. The remaining studies are interesting at worthy of publication is not already found in the literature.

Author Response

Thank you so much for your useful comments, below you will find our answer.

For the introduction several changes have been carried out on the text, in order to make it less complicated and clearer. For example, longer sentences such as on lines 72-76 have been reduced to simpler ones and a main goal for the study has been explicitly stated. Also, as requested more details on the identification of the components of the crude extracts have been provided, with a clearer reference to our past work.

Beside the introductive sentence of lines 58-60, clearly stating our previous study on the composition of the studied crude extract we have also linked this work to our previous one in the conclusions (lines 444-446).

The cytotoxicity reported for the CHCl3/MeOH extract, which is the subject of this study, still falls into the definition of mildly cytotoxic extract, has established by the American NCI guidelines. This said, the activity of fractions and p-CA is clearly low and definitely suggests the presence of a synergistic activity of the components of the extract which has been now added more clearly to the discussion. Nevertheless, p-CA could still be interest in terms of platform for the development of more active drugs and its direct interaction with protein kinases retains its interest.

ROS production was not studied since the production of these species (like H2O2) by polyphenols is already a known process and so, we aimed to find alternative mechanisms of action of p-CA by microarray analysis of the gene expression of PC3 cells treated with the phenolic acid.

We hope our answer clarifies your doubts.

Reviewer 2 Report

Review for molecules-1022256 and manuscript entitled: “Cytotoxic active principles from Hechtia glomerata extracts and p-coumaric acid as MAPKs inhibitor” In my opinion the manuscript can be accepted for publication after improvements that are detailed below.

Detailed suggestions:

  1. The hexane extract showed interesting activity against Hep3B, MCF7 and HepG2. In order to understand their compositions, it is recommended to analyze them by GC-MS.
  2. Pooled fraction 4 was not as active as the CHCl3/MeOH extract on both prostate and breast cancer cell lines. Please explain its possible reasons.

Author Response

Thank you very much for your comments and suggestions, below you will find our answer.

The GC-MS analysis of the hexane extract was already studied on one of our precedent works, as reported in the introduction (lines 58-60). But, to avoid confusion it was decided to eliminate those findings from the manuscript focusing only on the studied cell lines sensitive to the CHCl3/MeOH extract which is pivotal in our study.

The lower activity of fraction 4 with respect to the whole extract suggests that there is a synergistic activity between the active components of the CHCl3/MeOH extract. To make this clear this hypothesis was add to the discussion of the manuscript on line 163.

We hope this answers your doubts.

Reviewer 3 Report

Authors described para coumaric acid as an active component and its effect on up or down regulation in PC3 cells.

There is no clear reason to select this compound even this compound is not potent.

Also, no isolation work in the present study was done. Bioactivity-guided separation may be required.

Author Response

Thank you for your comments, below you will find our answer.

p-CA was selected for our study because of the previous studies on its biological effects and because it was found among the constituents of the active fraction of the plant. We understand that the reported activity of this compound may not be remarkable, but this study is still interesting since it gives more hypothesis on the use of this compound as a platform for the developing of new and more potent kinase inhibitors. Furthermore, its effects on the expression of genes related to chemotherapy resistance makes it appealing as re-sensitizing agent.

In this case the study reveals the constituents of the active fraction obtained from the CHCl3/MeOH extract of the studied plant, by HPLC-MS analysis. Of course, to proof some of our hypothesis isolation work and further analysis will be necessary, this is why this study has now been clarified as preliminary.

We hope this answered your doubts.

Reviewer 4 Report

The authors describe the early stage of a bioguided fractionation study of Hechtia glomerata. It is limited though to the very first fractionation as the study then relies on LC-MSn analysis of the most active fraction to list the secondary metabolites it can contain.  Despite its weak activity against tumoral cell lines, authors focused on para-coumaric acid, studied its potential mode of action through microarray assay. They identified a potential target (MAPK8) and studied its potential binding mode. Authors should be careful with their statements and clearly mention that their results led to strong hypothesis when it relies on in silico study, literature comparison and has to be confirmed with in vitro assays with pure compounds. On another hand, they can be really conclusive when it relies on the results of microarray assays.

Please find below some selected comments. Others available in the pdf file also need to be answered and changes have to be made before going through the next step of the publication process. This paper needs these major revisions to be done to be further considered. 

From a general point of view, some sentences are quite long and could be divided into shorter sentences to improve the reading and to help the reader to understand the authors ideas.

line 2 : in the title authors should change principles for fractions as the only pure compound evaluated, p-CA, is weakly active. Authors themselves conclude that the next step is to further fractionate the most active fraction to isolate and characterize the potentially active secondary metabolites (lines 261-262).

line 22 : This classical approach is limited to one step. Therefore the use of "bioguided fractionation" has to clearly show that it is a preliminary fractionation. even though the LC-MS study allowed to list the consitituents of the fraction 4, authors do not have in hand all these compounds purified from this matrix.

lines 57-59 : it is difficult to believe that the authors sincerely aimed at studying thoroughly "one" compound before this work. Authors should rather mention "the most cytotoxic or the major compound". It should not be forgotten that cytotoxicity of the extract can rely on a mixture of several compounds as observed in traditional medicine uses.

line 72 : Why do the authors directly start with the cytoxicity of the extracts? they should give some results dealing with the extraction they ran, the yields they obtained with each solvents they used...

line 73 : The authors should focus on PC3 and MCF7 assays as their selection of extracts will rely only on the results associated with these two cell lines. Thus they should remove their weak explanation not to explore hexane extract (lines 87, 88, 89).

line 76 : Authors should explain the use of such mixture. Extraction is usually done with a gradient of polarity from hexane to MeOH...

line 86 : Do the authors mean that the ethnomedical use of the plant refer to a chloroform/methanol extract?

line 130 : Authors should give/estimate the percentage of p-CA in fraction 4 and also mention if it is found in other pooled fractions.

line 188 : add a reference for these derivatives. 10.1016/j.mcn.2011.12.005
10.1073/pnas.0805677105

line 220 : Authors should mention here the pdb code of this structure. Why did the authors use this particular pdb accession with such resolution?

line 288 : what were the weights of these pooled fractions?

line 418 : Based on preliminary publication from the same authors (reference 11), it sounds difficult to mention "first study of this kind". Authors have to change this for "completed the data obtained from previous studies of the extracts of this plant."

line 442 : Authors should check the references section (among other remarks please check italic for plant names, titles sometimes contain upper cases for all the words, extra spaces ref 18, name ref 19, extra characters ref 25, journal abreviations ref 40 and 42,...)

line 480-482 : Reference 14 is not a validated paper and should be removed from the references section.

line 495-497 : Reference 19 is not a reference, just an abstract of conference extracted from ResearchGate!!! Results are probably associated to reference 8...

Sincerely yours.

Author Response

We wish to thank the reviewer for its comments, we carefully reviewed all the grammatical errors and changes that were pointed out. Following is our answer to the comments.

Lines 2 and 22: As requested, the title was changed by replacing “principles” for “fraction, also it was specified in the abstract that this study a preliminary fractionation.

Line 32-34: Not straightforward links were eliminated to avoid confusion. Also, with the highlighted sentence the authors meant literally that the found binding site of p-CA on the studied proteins was in fact the same site occupied by some reported inhibitors. This was found by aligning the already reported pdb files of the selected proteins (crystallized bound to their inhibitors) and the Molecular Docking poses which presented the best score. The possibility of other allosteric binding sites was not contemplated.

Line 42-43 As suggested “the latter” was eliminated.

Lines 55 and 57-59: Instead saying “one” compound, it was decided to say, “one of the most cytotoxic”, because among the isolated molecules p-CA was the most stable and cytotoxic. Also, references 10 and 11 were cited as suggested.

Line 61 and 69-70: The phrases were changed to be more clear and shorter ones, and repetitions were also eliminated. Also, a main goal for the study was added.

Line 72: A paragraph was added in the Results and discussion section, before the cytotoxicity results, to clarify the results of the extracts preparation.

Line 73: As suggested, all details relative to the other tested cell lines have been eliminated to focus the manuscript only on the sensitive cell lines and therefore deleting also any unneeded justification.

Line 76: The explanation for the use of the CHCl3/MeOH was added in the new paragraph reporting results of the solvent extraction, which is to widen the range of metabolites to be extracted and maximizing the extraction efficiency.

Line 78: A proper reference was added.

line 86-87: No previous activity of the plant was linked to the CHCl3/MeOH extract, but the reported ethnomedical use of the plant gave reason for the further investigation of the active extract along with its complex composition. Also, the sentence was rebuilt to adjust the addressed unclarities.

Line 98: The paragraph now starts with some experimental details as suggested: such as HPLC column, gradient of the mobile phase and MS analyzers employed in the compounds’ detection.

Line 107: The sentence was changed to be more concrete and a table of all the mentioned compounds was added at the end of the paragraph.

line 115: Only the cell lines of interest were reported. The sentence was rephrased to show the lowest concentration reported also, the unit of measure of the IC50 were changed back to µM as requested; and the reported cytotoxicity was compared to the positive control. For the values in pg/mL: none of the references we mention in the text report such low values.

Lines 124-125: As suggested the sentence was changed, explaining the necessity of further study to actually proof the conclusions.

Line 130: An estimate of the percentage of p-CA in fraction 4 extrapolated from table 3 was reported as well as its presence in other fractions, discussing on these bases the observed cytotoxicity.

Lines 134-135: It was specified that cellular assays are needed to proof our conclusion and the probable synergistic action of the fraction components was mentioned.

Line 140: A discussion between the activity of p-CA and other natural polyphenols was added to in this same paragraph to better justify its study.

Lines 188 and 220: The suggested references were added for the cited inhibitors. Also, the pdb code was provided and as mentioned in the paragraph, this particular pdb was select because it contained the structure of the protein with its inhibitor within its binding site. This was thought to give the best approximation of how the protein would change its conformation in the presence of its inhibitor and therefore give a more realistic molecular docking score. Also, this pdb resulted also complete structurally which eased its processing and hence the computational cost.

line 191: the book is appropriate for the discussion of inhibitors since in the section describing cell signalization pathways, there are many references to protein kinases inhibitors. More in specific, at the end of the paragraph which explains the JNK pathway are found the information cited in the manuscript.

line 251: figure 2 was changed in order to apply the requested color code for the arrows.

Line 288: as suggested the weights for all fractions were added to table 2.

Line 418: as suggested the conclusions were changed by implying a continuity between the previous studies and the present.

Lines 442: the references were corrected based on the reported errors, the extra spaces and characters were eliminated, all authors names have capital letters, and all scientific names and Latin expressions were changed in italic.

Lines 480-482 and 495-497: all the incorrected or non-valid references were changed for valid one, in specific the congress abstract was changed for the actual published work and the and the non-validated article was changed for a valid one.

lines 556 and 563: The Standard Abbreviation (ISO4) of Bioorganic and Medicinal Chemistry Letters is “Bioorganic Med. Chem. Lett.” meanwhile, for Free Radical Biology and Medicine is “Free Radic. Biol. Med.”. These abbreviations were also found to be used in the NCBI database therefore, the references were not changed.

Round 2

Reviewer 2 Report

The manuscript can be accepted for publication.

Author Response

Dear Reviewer thank you for your valuable revision and comments

Introduction, research design as well as the conclusions were improved as it shown in the corrected version.

The manuscritp was edited by a native english professional

We hope you are agree with the corrections that we made to the manuscript

Best regards

MRCC 

Reviewer 3 Report

Authors have to isolate the active compounds or reference markers.

Author Response

Thank you very much for your comments, below you will find our answer.

At the moment we are unable to perform any compound isolation work on the active fractions since they were already processed in the isolation of the major compounds already reported in our previous work.

Therefore, new vegetal material recollected in the same place and time of the year as the first batch, will be needed. This in order to obtain a sufficiently high quantity of new CHCl3/MeOH extract and fractions to employ in the bio-directed phytochemical study for the isolation of its cytotoxic active principles.

We hope this answered your doubts.

Sincerely,